# Coniferous Trees as Bioinspiration for Designing Long Reinforced Prestressed Concrete Columns

**DOI:** 10.3390/biomimetics9030165

**Published:** 2024-03-07

**Authors:** Traian-Nicu Toader, Călin G.-R. Mircea, Alina M. Truta, Horia Constantinescu

**Affiliations:** 1Department of Structures, Faculty of Civil Engineering, Technical University of Cluj-Napoca, Str. Constantin Daicoviciu 15, 400020 Cluj-Napoca, Romania; calin.mircea@dst.utcluj.ro (C.G.-R.M.); horia.constantinescu@dst.utcluj.ro (H.C.); 2Department of Forestry, University of Agricultural Sciences and Veterinary Medicine, Str. Calea Mănăștur 3–5, 400372 Cluj-Napoca, Romania; alina.truta@usamvcluj.ro

**Keywords:** biomimetic concrete column, Norway spruce, prestressed reinforced concrete, biomechanics, energy dissipation, walnut shell

## Abstract

This article contains the results of identifying the potential of coniferous trees to act as bioinspiration for the structural design of columns in single-story warehouses subjected to high wind velocity and severe seismic action. This study starts by analyzing the biomechanics of coniferous trees, continues with an abstraction of the relevant features, and ends with the transfer of a design methodology for long reinforced and prestressed concrete columns. To verify the applicability and validity of the mathematical relationships extracted from the bibliographic study to characterize the biomechanics of coniferous trees, a study site is conducted for Norway spruce trees felled by the wind in the Bilbor area. The design methodology for long reinforced and prestressed concrete columns bioinspired by the Norway spruce trees is experimentally validated using two case studies. The first case study deals with the effect of centric prestressing on long concrete columns, and the second on the influence of the walnut shell powder on the adhesion of the reinforcement in concrete. The case studies presented aim to transfer some characteristics from trees to reinforced concrete to improve the performance of long columns under horizontal forces. The results obtained indicate a good approximation of the trees’ structural behavior for this site and for ones investigated by other researchers in different forests.

## 1. Introduction

People’s needs for sheltering new technological processes in factories, storing material goods in modern logistics centers, and building mega shopping centers have been materialized through projects of single-story constructions with significant heights on sites with severe seismic actions [1]. The need for large free heights of 10 m and above results from the organization of technological flows and the good functionality of the building, while the site is chosen mainly according to the development of the infrastructure and the workforce available for future economic activity [2]. Considering the above criteria, it is observed that the free height of the columns that make up these structures is almost impossible to reduce, thus the structural engineer is responsible for finding solutions suitable for the design theme, which must also be economically competitive and in line with sustainable development.

In nature, materials have specifications starting from the atom level, such as spider silk, wood, shells, etc. People have imitated nature by creating new synthetic materials such as cement materials (e.g., concrete) [3,4] and alloys (e.g., steel), sometimes starting at the atom level, such as Nylon and Kevlar [5].

Columns are crucial structural elements in a load-bearing structure. They have the role of transferring the loads applied at roof level, on the facades, and those on their height to the foundations. In single-story warehouses (Figure 1), the columns are often subjected to high eccentric compression and shear force, identical to trees. From a mechanical point of view, the trunks of trees have the role of transferring the loads applied to the crown and trunk (own weight of the branches and trunk, precipitations, and wind action) to the root system [6,7,8,9]. For both columns and trees, the static scheme is that of a vertical cantilever embedded at the base. As the length of the cantilever increases, it is necessary to increase the bending stiffness to avoid excessively large lateral displacements leading to loss of stability; therefore, it is necessary to increase the resisting bending moment to be able to take up the demanding bending moment, otherwise the cantilever would break in the most severely stressed cross-section. The bending stiffness can be expressed as E·I, where E is a mechanical characteristic of the material (longitudinal modulus of elasticity) and I is a geometric characteristic of the section (moment of inertia or, more specifically, second moment of area). Therefore, to obtain a higher bending stiffness, we need to increase at least one of the terms. If we keep the materials used, it follows that increasing the stiffness can only be achieved by increasing the dimensions of the cross-section.

Nowadays, the most used and financially accessible building materials are concrete and steel. Concrete has physical (high resistance to fire and aggressive environment, medium specific weight, low thermal expansion, etc.) and mechanical properties (sufficient compressive strength, favorable elasticity mode, etc.) that make it recommendable for widespread use in load-bearing structures. However, the low tensile strength of concrete is an impediment, involving its association with other reinforcement materials (with tensile strength 100 to 1000 times that of concrete), resulting in a composite material, i.e., reinforced concrete [10]. An excellent reinforcement in concrete is steel in the form of ribbed bars. Steel has a thermal expansion compatible with that of concrete and good adhesion when embedded. When reinforced concrete structures exhibit deformations above the serviceability limits, or when the goal is to reduce the dimensions and the elements’ own weight, active reinforcement (pre-tensioned or post-tensioned) can be used in association with the concrete, thus resulting in a composite material like the first but with superior mechanical performance, i.e., prestressed concrete. In the modern era, engineering solutions for the reinforcement (using passive reinforcement) of reinforced concrete have been studied and then successfully implemented by engineers and professors such as Emil Mörsch and Fritz Leonhardt [11,12,13,14,15,16]. Regarding prestressed concrete, prestressing engineering solutions were studied and successfully applied in construction by the pioneering French builder Freyssinet and immediately continued by his successors [17,18,19]. Since the beginning of the modern use of reinforced concrete, the strut-and-tie model and the stress-field model have been used as design principles, with reinforcements positioned inside the concrete member where tensile stresses would exceed the tensile strength of the concrete.

The purpose of this article is to identify the relevant mechanical, physical, dynamic, and morphological properties to verify the potential of coniferous trees (Figure 2) as a biological role model in the design of long reinforced and prestressed concrete columns. Being aware of existing differences, this article is focused on the similarities that can improve the design of this type of column and its foundations within single-story warehouses.

In this article, bioinspiration is oriented towards a direct and pragmatic applicability in structural engineering, and, through biomimetics, an attempt is made to decipher the natural design strategy of coniferous forest trees transferring to long prestressed reinforced concrete columns and their foundations. The particularities of the trees are analyzed through the lens of biomechanics, considering as studied parameters statics, dynamics, morphology, rooting, and energy dissipation.

## 2. Analysis and Abstraction of the Biological Model

### 2.1. Biomimetics and Trees

Biology has a large potential to generate structural solutions for today’s reinforced concrete constructions [20,21]. The present research aims to clarify how tall forest trees have an amazing ability to withstand high horizontal forces. The form–structure–model relationship is investigated by abstracting and adapting bioinspired load-bearing structures. Biological models are processed to understand the natural response and the natural balance within the structure. We must notice that such an approach does not explain the entire complexity of the natural biological role model but tries to discover the mechanical fundamentals through complex modeling of innovative reinforced and prestressed concrete members. Figure 3 shows the relationship between biology and technical implementation in case studies comprising analysis, abstraction, and implementation [22].

### 2.2. Dynamics and Damping

**Dynamics.** Biomechanics studies trees as mechanical objects [23] using principles from engineering and physics to understand the structural properties of trees and how they interact with the environment. Tree growth rate is directly influenced by physiological aspects, especially those affecting photosynthesis and water transport [24]. But whether they are optimal or not, the size and shape of the tree is limited by biomechanical constraints [25]. Tree wood, as well as most plant materials, is viscoelastic because its mechanical properties are both elastic and viscous [26]. These properties involve a non-linear behavior [27,28], and, when mechanically stressed, live wood does not fit into a current mechanical model, as, for example, it does for concrete and steel. Thus, it is important to acknowledge the limitations of attempts to characterize and quantify trees using exact parameter values, and to recognize the failures when theory and reality do not coincide [29]. In addition, biological materials change their properties as trees grow and age [25,30,31,32,33,34,35], thus making the dynamic response difficult to predict.

**Frequency.** The frequency of swinging trees has been the subject of numerous studies and proposed relationships for the dominant frequencies of trunks [36,37,38,39,40,41,42,43,44]. Figure 4 shows a synthesis of the natural frequency determined on coniferous trees. Based on the cantilever beam model, Blevins [45] proposed the following relation for the natural vibration frequency:(1)$$f_{n}=\frac{\lambda^{2}}{2\cdot \pi \cdot L^{2}}\cdot \sqrt{\frac{E\cdot I_{0}}{\rho \cdot A_{0}}}$$
where λ is a dimensionless parameter depending on the variations of the tree properties (e.g., shape, mass, and stiffness distribution), L is the height of the trunk, E is the elasticity modulus, I_0_ is the second moment o area at the base (the basal area moment of inertia), ρ is the density of the material, and A_0_ is the cross-sectional area at the base of the cantilever beam (trunk).

Furthermore, Scannell [46], Petty, and Swain [47] quantified the influence of the branches in decreasing the frequencies of the amplitude of the trunk swings. On the other hand, the empirical relationship proposed by Moore and Maguire [36] was validated for about 600 coniferous trees of different heights, diameters, and species:(2)$$f_{n}=0.0948+3.4317\cdot \frac{DBH}{L^{2}}-0.7765\cdot I_{p}\cdot \frac{DBH}{L^{2}}$$
where L is the height of the trunk, DBH is the diameter of the trunk measured at chest level (approximately 1.35 m above the ground), and I_p_ is a parameter equal to 1 if the genus is Pinus and 0 otherwise.

**Damping.** While the natural frequency of a tree is relatively easy to quantify and overall measure, the quantification of damping is much more complex [28]. Tree damping includes several components, e.g., (a) aerodynamic damping, (b) internal or viscoelastic damping, (c) mass damping, (d) damping by root–soil interaction, and (e) collision of branches or other crowns [28,48]. Figure 4 summarizes the frequency and damping values of about 900 coniferous trees of different sizes and masses, calculated and published in [36,37,38,39,40,41,49,50,51,52]. However, the fraction of critical damping ξ (damping ratio) can vary greatly. In particular cases, it was observed that, due to the very high damping by the oscillations of branches (c) and the collision between branches (e), the trees were almost critically damped (ξ ≈ 100%), returning to their original position (rest position) after only one or two cycles of free vibration [49].

In the experimental and numerical studies of [49,53], it was observed that by removing the branches, the natural frequency of the tree changed and the damping started to decrease, but only after about 80% of the branches were cut (Appendix A).

**Figure 4 biomimetics-09-00165-f004:**
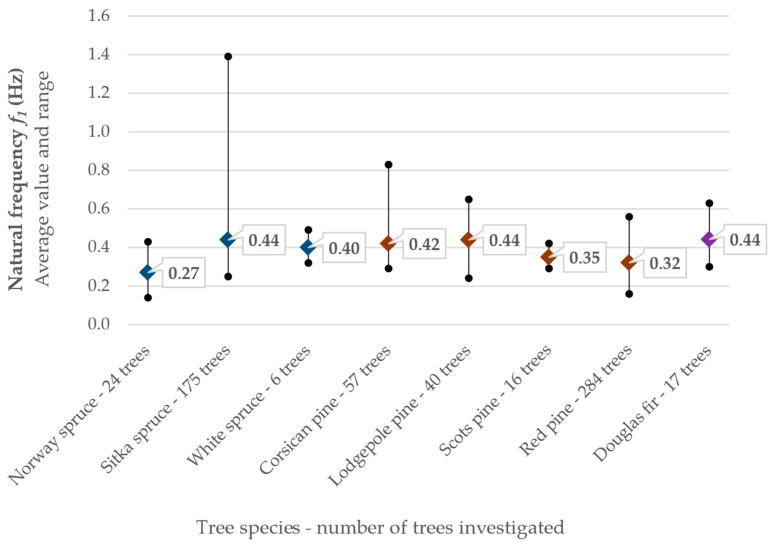
Natural frequency and damping ratio of conifer trees. Usage mean values are presented along with ranges, using input data from [36,37,38,39,40,41,49,50,54,55].

### 2.3. Morphology and Deformation Capacity

**Morphology and material properties.** The dynamic behavior of the trees is dominated by the size and morphology of the specimen [24]. Small morphological changes can lead to spectacular differences in the dynamic behavior of the trees [28,56]. Furthermore, different shapes of the trees are characterized by distinctive scattering of the material density, stiffness, and mass distributions [35]. Thus, the shape of the tree has more significant influence than the properties of the intrinsic material [28]. The response of the tree to the strong wind acting on the dominant direction results in lower heights of the tree [28], higher material density of the trunk on the extreme compressed fiber [57], more developed roots [58,59], larger diameter of the branches, and crowns in a flag shape [60]. At the same time, in response to wind action, changes are observed in the structure of the cell walls and in the properties of the wood [61]. Natural wood from woody plants mainly contains about 75 ÷ 80% cellulose and lignin. At the level of the cambium, the trees produce reaction wood so that the cross-section can withstand high mechanical stresses. In gymnosperms (i.e., coniferous trees), reaction wood is a compression wood type, having a high lignin content and lower cellulose content [62]. Compression wood has eccentric annual growth rings and a higher density than normal wood. Compression wood is found in the compressed fiber of trunks and branches [63], in contrast with most angiosperms, where the reaction wood is called tension wood, having an increased cellulose content of up to 60% [62,64]. Tension wood is found in the stretched fiber of trunks and branches. In porous woody angiosperms, hardwood fibers of tension wood have also been observed to produce a gelatinous cell wall layer. This gelatinous layer allows the tension wood to have greater elongations than the surrounding wood [61,65].

The Norway spruce has a macroscopic structure of a cellular solid type consisting of parallel tubes, called wood cells, that have a hierarchical internal structure [66]. The stiffness and the strength of wood are much higher along the longitudinal axis compared with directions perpendicular to it [67]. The cell wall is a fiber composite made of cellulose microfibrils embedded into a matrix of hemicelluloses and lignin (Figure 5) [68]. The matrix allows relatively large shear deformation between neighboring fibrils [69]. Cellulose fibrils wrap the tube-shaped woody cells at an angle called the cellulose microfibrils angle, γ, which varies between 0° and 90° [66].

**Plastic deformation in wood.** Wood is a composite material, with wood cells having a rather complicated deformation behavior, especially in large deformations [66]. In general, a typical σ-ε (normal stress-specific longitudinal strain) curve for ductile materials has an elastic slope for small strains, followed by an elastoplastic or even plastic slope for large strains. In studies on spruce, it was observed that the angle of the microfibrils decreases when the strain increases [69,70], the microfibrils reacting like a spring. At the trunk level in the stretched fiber, bending deformations produce a reduction of the angle of the microfibrils (from γ to γ’) that occurs simultaneously with a shearing of the matrix between the fibrils (Figure 5). The fibrils can be considered inextensible, and the deformation takes place because of the sliding of the cellulose fibrils against each other thanks to the property of the matrix between the fibrils (Figure 5). Undeformed cellulose fibrils take up most of the load, while deformations are consumed by shearing the hemicellulose and lignin matrix. An important condition for this deformation mechanism to happen is the existence of a strong bond between the matrix and the fibrils, there being a chemical compatibility between the hemicellulose and the fibrils (both being polyoses). Hemicellulose acts as an adhesive between the cellulose fibrils and allows sliding [66].

**Mechanochemical model for the deformation of hemicellulose in cell walls of coniferous wood.** At the contact between hemicellulose and cellulose fibrils, a hydrogel-type matrix is formed, and it is assumed that, when the fibrils are subjected to axial stress at the fibril–matrix interface above a certain limit value of the unit shear stress (τ), the matrix shears and flows by opening and reforming hydrogen bonds. The mechanical response of the hemicellulose and lignin matrix is represented as a characteristic curve τ-η (shear stress–strain, Figure 5) [66,69,71]. 

**Young’s modulus.** To be able to perform a modal analysis of the tree and to determine its eigenmodes and their corresponding frequency, the Euler–Bernoulli formulation may be used. In this regard, it is necessary to know the longitudinal elasticity modulus of the material and the geometric characteristics of the equivalent cantilevering beam. For about 650 trees, averages of these quantities are summarized in [72] and shown in Table 1. Given that the cellulose microfibrils remain mainly undeformed, the Young’s modulus of the material is mainly reported on their stiffness.

### 2.4. The Root System: Reaction Forces (Tree–Ground Anchorage Forces)

The root system can be regarded as a foundation system of the superstructure. However, in contrast, the root system brings together the root biomass with the related soil. Moreover, it is a living organism that grows together with the crown and the trunk. Its mechanical performance is given by the roots’ properties, the bonded soil, and the size of the system, which is variable as the cross-section decreases with the depth [9]. Figure 6 explains the four mechanisms through which the root–soil connection gives resistance to the tree base.

When a horizontal force acts on the tree trunk, the weight of the roots and their associated soil help to weigh down the root–soil plate, which is the first component. The soil under and around the plate is broken during uprooting (Figure 7); consequently, the soil’s tensile strength contributes to the load-bearing capacity of the foundation (the second component). The third component is the tensile strength of the roots parallel to the direction of action of the horizontal force. And the fourth is the bending resistance of the roots and the soil around the plastic zone (area b in Figure 6) [73].

The length of the root network increases with age and can reach kilometers or even tens of kilometers [74]. The anchoring performance of the system is given by the roots that carry the tensile stresses and the adjacent soil sustaining the compression stresses. The reaction forces at the root–soil plate level depend on the following factors [9]:Structure and mechanical properties of the roots;Spatial distribution and way of anchoring of the roots;The structure and physical/mechanical properties of the soil, of which moisture plays an essential role;The interaction between the roots and the surrounding soil.

The distribution of the reaction forces into the four components is very variable due to the highly complex structure [9]. However, considering a body of rotation as a biomechanical model with the main and lateral roots generally converging in a direction towards the theoretical fixation point O_1_ (Figure 8), it can be further considered that the directions of the reaction forces in the roots and their resultant forces, T (tension) and $C_{inf}+C_{lat}$ (compression), converge at the same point. It follows that, for the tree loaded with self-weight + external actions, in a stable equilibrium state, the tensile reaction forces in the roots act on one area of the contour surfaces, and the compression from the ground reinforced with roots acts on the rest. To quantify these tree–soil connection forces, a force distribution model on the contour surfaces can be adopted. In this context, Grudnicki [9] adopts the simplified linear distribution of the reaction forces on the contour surfaces, which also facilitates the determination of the resultant forces T, C_inf_, and C_lat_ (Figure 8).

### 2.5. Failure of Trees under Wind Action

Mayer states that no tree can withstand a violent storm and questions how to put into practice the results obtained from investigations on tree oscillations [76]. The failure mode of the trees under the dynamic action of wind is unknown to us because the real dynamic process has never been verified by field experiments on a natural scale in which all the relevant parameters have been monitored [77]. And the assumption that peak wind loading during a storm is the key factor causing degradation has never been verified in situ, and it is possible that a more important factor is root fatigue from several previous storms [77].

Studies on the impact of certain hurricanes on urban pines in Florida, USA show that failure by trunk breakage occurred mostly in Pinus elliottii (slash pine) trees—64% during Hurricane Jeanne—while the majority of Pinus clausa (sand pine) trees broke only during Hurricane Jeanne—71% [78]. Continuing with another hurricane (Ivan), here the primary failure mechanism was uprooting. Except for post-storm investigations in which the wind speed of tree failure was estimated [79], to date there are no scientific methods that can predict tree failure at a certain wind speed [24]. In Figure 9, the representative shapes of tree instability are sketched [7].

**Resistance to uprooting.** An approach for the assessment of uprooting resistance is to estimate the weight of the root–soil plate (foundation consisting of roots + associated soil) and then calculate the resisting bending moment at overturning [80]. The tree is uprooted if the resisting bending moment at overturning is exceeded, while this moment depends on the weight, depth, and diameter of the foundation and on the properties of the soil [43,73]. In addition to the weight of the foundation, there are other factors that can contribute to the stability of the tree foundation, such as the tensile strength of the roots parallel to the direction of the wind or the plasticizing strength of the roots and soil. Due to the complexity of the individual assessment of each individual factor, their input is introduced into Equation (3) as the coefficient A_rsw_ [80], based on tree pulling experiments from [81]. The A_rsw_ coefficient indicates the ratio between the weight of the root–soil plate and the total anchoring force of the roots in the soil [73,80].
(3)$$M_{R\_rs}=\frac{m_{rsplate}\cdot g\cdot h_{rsplate}}{A_{rsw}}$$
(4)$$h_{rsplate}=\frac{h_{rscone}}{3}$$
where M_R_rs_ is the resisting bending moment of the total root–soil plate anchorage (kN·m), m_rsplate_ is the fresh mass of the root–soil plate (kg), g is the gravitational acceleration (m·s^−2^), and h_rsplate_ is the mean depth (m) of the root–soil plate volume (Equation (4) and Figure 10). The contribution of the root–soil plate to the total anchoring force is considered 30% for Scots pine, 20% for Norway spruce, and 30% for birch, respectively [80].

**Resistance to stem breakage.** If we accept the hypothesis that when the tree trunk is bent, or any cross-section of the distribution of normal stresses varies linearly, then the maximum stress values at the edge fibers decrease to zero as we approach the neutral axis [80,82]. In this hypothesis, the critical section is considered at the height z = 1.35 m (breast height) measured from foot to top, the trunk diameter being equal to DBH (diameter at breast height) [80]. The trunk is considered to break when the maximum stress exceeds the flexural strength for green wood, f_fl,gw,stem_ [47,83,84], while the resisting bending moment to breaking the trunk (maximum turning moment a tree stem can withstand without breakage) can be calculated using Equation (5) [80,82,85].
(5)$$M_{R\_stem}=\frac{\pi \cdot {DBH}^{3}}{32}\cdot f_{fl,gw,stem}$$

Figure 11 shows values of resistance to overturning by uprooting and by breakage of the trunk for several Scots pine trees [86]. Analyzing the results, one can see that all trees loaded with self-weight and a lateral force failed by overturning with uprooting, the trunk bending resistance being almost four times higher than the uprooting resistance. The following data are considered in the calculations: modulus of elasticity 7000 MPa, flexural (bending) strength of the stem 32 MPa (=70% of breaking stress), fresh density of soil 1700 kg/m^3^, tree species Scots pine [86].

**Figure 10 biomimetics-09-00165-f010:**
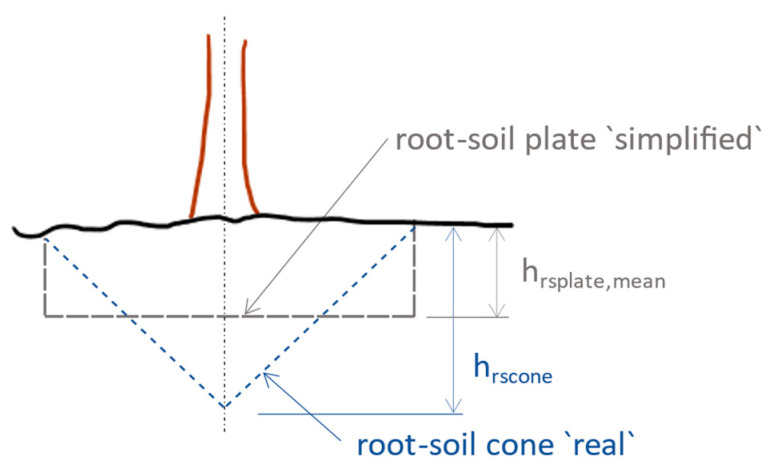
Root–soil plate at tree uprooting; cone and circular plate. A sketch of the concept described in [86].

### 2.6. The Biomechanical Model of Conifers

In terms of stability, a tree can be divided [9] into:Elevation, consisting of stump, trunk, and crown;Foundation, consisting of roots and related soil, called the root–soil system.

Modeling the trunk as a beam element of variable cross-section along the length of the shaft has been shown to provide valid results in determining the natural period of vibration [72]. This solution allows the biomechanical model to consider the degree of flexibility/fixation of the trunk in the foundation.

So, a first simplified option is to consider the trunk as a cantilevered vertical beam, fully fixed at the base and with a bending stiffness adjusted according to the effective stiffness of the trunk–root–soil system. By idealization, the elevation is considered fixed in the foundation at ground level, and the root system is considered to be a reinforced foundation created by the tree [9]. The appropriate body of rotation as a biomechanical model for the root–soil system of the spruce is a cylinder or a truncated cone of shallow depth (Figure 10). To establish the distribution of self-weight and external loads, one starts from the contour curves of the spindle and the crown; in this field, there are mathematical relationships published in specialized works of dendrometry [9].

The second option consists of joining the trunk–root–soil system, consisting of the trunk modeled as a beam with a certain spring and rotation stiffness. There is a lot of information about the dynamics of fully fixed cantilevers but much less about cantilevers with semi-rigid joints. The latter situation is encountered in structural engineering, especially in the design of columns with foundations placed on soft soils. For a root–soil system of circular shape in the plane, the rotational stiffness equation can be written as for a circular plate placed on deformable soil, i.e., the Winkler foundation model (Winkler bedding coefficient/soil spring constant) [72,87,88]:(6)$$K=\frac{G_{rs}\cdot {d_{rsplate}}^{4}}{6\cdot (1-\nu_{rs-s})}$$
where G_rs_ is the shear modulus of the root–soil system, d_rsplate_ is the diameter of the circular foundation (approximated as a mean value), and ν_rs-s_ is Poisson’s ratio of the root–soil system. This calculation approach is specific to building foundations, but here G and ν refer to the behavior of the root–soil composite material in the root–soil system.

**Degree of fixation of the trunk in the foundation.** Knowing the degree of fixation of the trunk in the root–soil system helps us to quantify and distinguish the flexibility of the root–soil system from stem flexibility in living trees. The stiffness of the root anchorage is influenced by the modulus of elasticity, the cross-sectional area, and the architecture of the roots, and by the physical and mechanical characteristics of the soil. Mathematically, root–soil system stiffness can be quantified using an elastic spring constant/stiffness coefficient for root–soil rotational stiffness. Thus, in [89], Equation (7) is written for the secant stiffness, k_root_, where M_E_stem_ is the bending moment calculated at the base of the trunk, Φ_i_ is the rotation due to the pulling force, and Φ_0_ is the initial (existing) rotation before pulling test.
(7)$$\Phi_{i}=11.9\cdot {\left({DBH}^{2}\cdot H\right)}^{-0.53}$$
where Φ_i_ is expressed in degrees [°].

Regarding the spruce trees presented in [75], having the dimensions DBH_A_ = 69 cm and H_A_ = 39 m and DBH_B_ = 16 cm and H_B_ = 16 m, it was observed that the maximum bending moment (M_A_ ≈ 900 kN·m) for the one with the thicker trunk was achieved for a rotation Φ_A_ ≈ 3.5°, while, for the spruce with a thinner trunk, the maximum bending moment (M_B_≈11 kN·m) was approx. 80 times smaller but reached a rotation 10 times larger (Φ_B_ ≈ 26°). In [87], it was noted that root rotation contributed between 5 and 15% to the total flexibility of Sitka spruce trees.

### 2.7. Study Site

In Spring 2020, many coniferous trees (Picea abies) were downed by strong winds in the Călimani Mountains (Romania) (Figure 12). On this site, we conducted a study on an area of about 1 ha, with mainly Norway spruce vegetation (47°09′ N, 25°48′ E, ≈1125 m altitude), the age being 80 years. The average annual temperature is 5.4 °C The annual amount of annual precipitation is 926 mm at the nearest meteorological station. The soil type has been classified typical Districambosol, and the texture is brown acid. The slope of the land for the measured trees is 25 degrees, facing East.

The forest was affected by strong winds, with many trees being uprooted, broken, or left leaning (Appendix A). From these, we selected 30 specimens of uprooted trees, ensuring that they had close values for diameter, height, and crown size. We measured the diameter of the trunk, the length of the stem, and the dimensions of the root system (thickness, diameters, depth, (Figure 13 and Table 2)).

The calculations made based on field investigation (Table 2) show that the M_R_stem_/M_R_rs_ (-) average ratio was 4.9 (standard deviation 2.1, with range of values between 2.8 and 14.5). The natural frequency calculated using Equation (1) and verified with (2), shows that all trees studied had almost identical dynamic characteristics, the frequency being equal to 0.10 Hz. This confirms the hypothesis that trees in the same stand have equal natural frequencies. The two calculated parameters, the M_R_stem_/M_R_rs_ ratio and the fundamental frequency (natural frequency), fall within the average values obtained from the previously cited studies. As a result, the computational relationships and assumptions used for the biomechanical model confirm an adequate abstraction of the biological model.

## 3. Design Methodology Transfer from Coniferous Trees to Load-Bearing Structures

In general, the design of a structure starts from an architectural concept, most of the time aesthetically motivated but not always with well-defined ideas about methods of realization and performance in relation to its function and environmental conditions. Contemporary structural engineers use general design prescriptions, building statics, seismic analysis, and other such design tools to produce an initial design (structural concept). The structural concept may be refined through an iterative process until certain prescriptive conditions are met. Questions regarding the suitability of shape, building materials, construction methods, performance objectives, cost optimization, etc., are sometimes addressed from the beginning of the design process, especially by experienced and dedicated designers. But, at other times, such questions are addressed too late to add value to the project [5].

### 3.1. Steps from Biomimetics to Know-How Transfer

The first step is to identify and note the characteristics of coniferous forest trees that may be applicable and associate them with the design and construction of the single-story frame structures (fully fixed columns at the base and beams with hinges at the ends). Although there are many differences between biology and structural engineering, in this study, the focus is on the similarities that can help improve the design of tall single-story structures through bioinspiration.

In step two, affinity conditions are stated based on the in-depth study from step one.

And in the third step, experimental tests are used to verify the efficiency of the mimicked property.

### 3.2. Step 1: Identifying the Structural Characteristics of Coniferous Forest Trees

According to the bibliographic study undertaken, the following structural characteristics of coniferous forest trees are relevant for the study of long reinforced and prestressed concrete columns [5,28,90,91,92,93,94,95,96,97]:Trees are three-dimensional structures statically determined;From the building statics’ perspective, trees are vertical cantilevers;In a tree, the values of internal forces due to its own weight are minimal in relation to the external forces caused by wind and/or snow;All the elements of a tree are made of the same material, but the chemical composition, density, and mechanical properties can vary, and the load-bearing capacity varies along the element depending on the size of the applying force in that cross-section;Trees are believed to have a minimum mass structure with elements optimized for function and shape;The lack of mechanical ductility of the trees is compensated by greater flexibility and damping;The average fraction of the critical damping, ξ, lies between 5% and 12.8%;Trees maintain relatively large lateral displacements in extreme wind conditions;Tree joints can have a quasi-plastic response to extreme loads;Tree joints are endowed with a higher tenacity than that of the trunk and branches;Trees are systems with several degrees of freedom and with high damping;Trees in the same stand, although they have different heights, have the same natural frequency;Due to the high damping capacity and the multitude of independently vibrating elements (leaves and branches), trees rarely enter resonance;Tree trunks are naturally prestressed in both directions, longitudinally and circumferentially;Tree roots are thus designed to deform and uplift to a certain extent to prevent permanent damage to the base of the trunk.

However, it is important to mention the structural characteristics of coniferous forest trees [36,37,38,39,41,49,50,51,72], which differ substantially and are difficult to implement in a single-story warehouse with long reinforced and prestressed concrete columns embedded at the base and hinged at the top. In the following, a Norway spruce tree in an arboretum is discussed in comparison with a column from a single-story warehouse, both of equal heights. The warehouse has a height of 10 m and a roof area per column of 300 m^2^, considering the columns as having a square cross-section of 60 × 60 cm (on a site with low seismicity) and 100 × 100 cm (on a site with high seismicity). The roof is made of main and secondary roof beams of prestressed reinforced concrete, on top of which is placed a light covering made of corrugated steel sheets and a heat-insulating layer (own weight of the roof skin is 45 kg/m^2^, technical load is 50 kg/m^2^, and snow load is 150 kg/m^2^):The slenderness of the spruce trunk is 5 times higher than that of the columns for low seismicity regions (a_g_ = 0.10 g) and 10 times higher than that of the columns for high seismicity regions (a_g_ = 0.30 g);The ratio between the weight of the crown and that of the stem (W_crown_/W_stem_ = 0.5) is 24 times smaller than the ratio of roof’s total loads (including self-weight) and column weight (W_roof_/W_column_ = 12) for areas with low seismicity and 8 times smaller (W_roof_/W_column_ = 4) for areas with high seismicity;The natural period of vibration for trees is between 10.0 and 2.0 s, while for a single-story warehouse it is between 2.3 and 0.7 s;The alternation of synchronous and asynchronous oscillations of the branches with the effect of dissipating the energy induced by wind or earthquake actions contrast with the movement of the roof beams connected to the column;The root system is a hybrid between a shallow and a deep foundation (with individual footing and ground anchors), while for reinforced concrete columns such a solution would be too expensive, being used especially for special structures such as towers for wind turbines. In general, nowadays, the common solutions used for the foundations of the columns in single-story warehouses are either a shallow foundation as individual footing or a deep foundation with individual footing sitting on piles.

## 4. Features of the Biological Role Model Meant to Be Abstracted and Later Transferred

Establishing the affinity conditions is the second step in the biomimetic design. If we start from the assumption that nature can be seen as a textbook for engineers and we consider the principles of structural design, to understand and transfer the design knowledge from a living tree to an engineering structure, for example, a single-story frame structure with columns as vertical cantilevers, the following affinity conditions must be met [5,98,99]:Structural applicability (geometric similarities and use and behavior of materials);Functional similarity (similar loading conditions and similar climatic actions);Similar structural response (behaving in the same way under comparable external actions);Cost efficient (being as profitable as possible in terms of material and energy consumption and production costs).

The chosen biological role model is the Norway spruce, and the features meant to be replicated in bioinspired long reinforced concrete columns with respect to the affinity criteria are:At the macro-level, longitudinal prestressing for gaining increased flexural stiffness and self-centering capacity (Figure 14). The technical implementation consists in using prestressed unbonded steel strands inside the reinforced concrete column;At the meso-level, viscoelastic damping through sliding of the cellulose fibrils with shearing of the hemicellulose and lignin matrix between them (Figure 15). The technical implementation is solved by greatly upscaling the fibrils (diameter of ≈3 nm) embedded into a matrix of hemicellulose and lignin and substituting them with steel strands (diameter of ≈9 mm) embodied in a concrete mixture with lignin and hemicellulose content. Thus, it targets a controlled bond slip of the steel strands when in tension or in compression.

## 5. Results and Discussion

Step 3 presents the design and testing of bioinspired structural concrete, first at the macro-level (a reinforced concrete column centrically prestressed) and second at the meso-level (characterization of reinforcement bond in concrete in the presence of lignin).

### 5.1. Experimental Study on the Influence of Centric Prestressing in Long Reinforced Concrete Columns

Chirițescu and Kiss carried out an experimental study on long prestressed and reinforced concrete columns in order to determine the influence of centric prestressing on the bending stiffness and energy dissipation capacity of concrete columns [100,101,102]. The study consisted of physical experiments and numerical simulations on columns with a cross-section of 250 × 250 mm (Figure 16), tested as cantilever elements with a length of 3.2 m and two forces concentrated at the top (W_ext_, the equivalent axial force due to roof’s self-weight of the warehouse; F_lat_, the equivalent lateral force caused by wind action and seismic action) (Figure 17). The materials used for series S01 were concrete C60/75 and reinforcing steel/passive reinforcement B500C, while series S02 is like S01 except that approx. 60% of the passive reinforcement area was replaced by prestressing steel/active reinforcement Y1860S7, so that the resisting bending moments for the two cross-sections were equal (Table 3). Column S02 was centrically prestressed in the longitudinal direction with a P_0_ force, resulting in a mean precompression stress of approx. 5.6 MPa.

As a result of the experimental tests in the laboratory, the following were found:For the same lateral force of 24 kN (approx. 80% of the failure force), the reinforced concrete column (S01) had an average lateral displacement of 426 mm (≈13.3% drift) compared with the prestressed reinforced concrete column (S02), which had an average lateral displacement of 289 mm (≈9.0% drift), which means an increase in stiffness of almost 50%;Section S02 was less ductile than S01, the energy dissipation capacity being reduced by about 40%. This was caused by the much lower ultimate elongation of the prestressing steel (2.2%) than that of the reinforcing steel (7.5%), collaborated with the uninterrupted adhesion (full bond) of the active reinforcement along the entire length of the column;The prestressed elements had self-centering capacity;The use of centrically prestressed reinforced concrete columns for single-story warehouses was efficient to reduce lateral displacements at the top of the building (through longitudinal prestressing, the bending stiffness associated with large bending moments was reduced, Figure 17). Cracking of the concrete occurred much later compared with reinforced concrete members without prestressing but, at the same time, a reduced value of the behavior factor must be considered in the seismic design depending on the displacement ductility factor (µ_δ_ = Δ_u_/Δ_c_) and on the real curvature ductility factor (µ_θ_ = ϕ_u_/ϕ_c_) of the cross-section.

### 5.2. Experimental Study on the Influence of the Walnut Shell on the Bond of the Reinforcement in Concrete

In this study, it was aimed to mimic the viscoelastic damping of coniferous trees during wind action. When their trunks bend and the cellulose microfibrils wrapped along the wood cells allow the trunk and branches to bend by shearing the hemicellulose and lignin matrix, the microfibrils remain inextensible. In the case of reinforced concrete columns, flexural deformation occurs because of cracking of the concrete and elongation of the steel reinforcement in the tension fiber of the cross-section. At high stresses, the reinforcement is stretched beyond the elastic limit and yields, with a strain at maximum force of minimum 7.5% (reinforcing steel of ductility class C). But, when prestressing steel is used (such as strands), the strain at maximum force is only 2.2% [100,104]. To increase the deformation capacity of the prestressed concrete elements (implicitly the energy dissipation capacity), new solutions are needed; in this case, a biomimetic approach is used. It is desired to increase the deformation capacity of concrete columns reinforced with steel strands not by elongation of the reinforcement but by controlled slipping of the strands. The technical implementation aims to obtain the hydrogel behavior of the strand-type reinforcement embedded in concrete. To obtain this result, an experimental study was carried out on the bond of seven-wire strand prestressing steel (Y1860S7), having a total diameter of 9 mm, centrically embedded in a standard concrete specimen (cube with side 100 mm, without transversal reinforcement) for a pull-out test (Figure 18 and Appendix A; Appendix A).

To establish the effect of lignin and hemicellulose on the reinforcement bond in concrete, a parametric study was carried out on concrete compositions of minimum class C30/37. The concrete class was chosen in accordance with the good execution practice of prestressed reinforced concrete elements (monolithic and prefabricated). For specimen V3, the concrete strength class obtained was C32/40, and for specimens V4, V5, and V6, C35/40 (Table 4). The parameters considered were lignin and hemicellulose, as lignosulfonate-based admixture (activated in the plasticizer, in V4), as additive of non-activated lignin and hemicellulose in the form of peanut shell powder (V5), and as additive of non-activated lignin and hemicellulose in the form of walnut shell powder (Figure 19) (V6). The dosage used was about 1% of the amount of binder (cement). The chemical compositions of peanut shell and walnut shell are noted in Table 5. The reference specimens (V3) were made of concrete without additives and admixtures.

It was observed that, for specimen V6 (with lignosulfonate-based plasticizer admixture + hemicellulose and lignin additive in the form of walnut shell powder), the pull-out force–slip curve for the strand embedded in concrete had the allure of the shear stress–strain curve for cellulose microfibrils wrapped in the hemicellulose and lignin matrix (Figure 20, Appendix A). The hydrogel-like mechanical behavior may be due to the walnut shell powder acting as a glue between strand and concrete. But, to demonstrate this action, microscopic studies should be carried out to find out exactly how the powder reacts in the composition of the concrete and the chemical bond at the concrete–strand interface. The concrete mixture used for the V6 specimens is noted in Table 6.

Although specimens V3, V4, and V5 had a maximum pull-out force approximately 40% higher than V6, none of them showed hydrogel behavior up to 10% strand slips in concrete. The results of the study from [105] indicate the potential of using steel strands as reinforcement in concrete elements with the addition of lignin and hemicellulose to achieve a viscoelastic damping. This behavior could be exploited in the direction of increasing the deformation capacity of prestressed concrete elements (with bonded strands), for example, for dissipative zones/potential plastic hinge regions of structural concrete members in structures designed for earthquake resistance.

## 6. Conclusions

Finally, we can write down the bioinspired means with the greatest potential to improve the structural performance of single-story buildings with long reinforced prestressed concrete columns [5,103,104,105,108,109,110,111,112,113,114]:Reaching viscoelastic damping is assured by using concrete with the addition of hemicellulose and lignin and/or some longitudinal reinforcements with an integrated friction mechanism along their length;Supplementary damping results from the interaction of the soil foundation, such as the controlled uplifting of the foundation (solution studied on onshore concrete towers for wind turbines);The fraction of critical damping in coniferous trees is in the same range of that in reinforced concrete structures, so ξ = 5% is the conventional value in the design rules [115,116], which can be increased up to 20% if additional dissipative elements are introduced;Centric longitudinal unbonded post-tensioning of the concrete columns increases bending stiffness and enables self-centering capacity;Designing the structures of neighboring buildings so that the natural frequency of each is equal (this way the seismic joints will have a minimum width), as revealed by the on-site measurements of the Norway spruce trees in the Bilbor region.

Further investigations to enhance the similarity between Norway spruce trees and prestressed reinforced concrete columns will include:Additional studies on transversal prestressing of columns to increase the degree of concrete core confinement and the rotation capacity at the base of the column;Experimental tests for evaluating the viscoelastic damping after cyclic loading and unloading;Durability and ageing tests on the special concrete mixture containing biomass (lignin and hemicelluloses);Checking the hydrogel behavior on cracked concrete samples;Full scale experiments on concrete columns integrating both mimicked features: self-centering ability and viscoelastic damping, with bonded and unbonded strands at the same time.

## Figures and Tables

**Figure 1 biomimetics-09-00165-f001:**
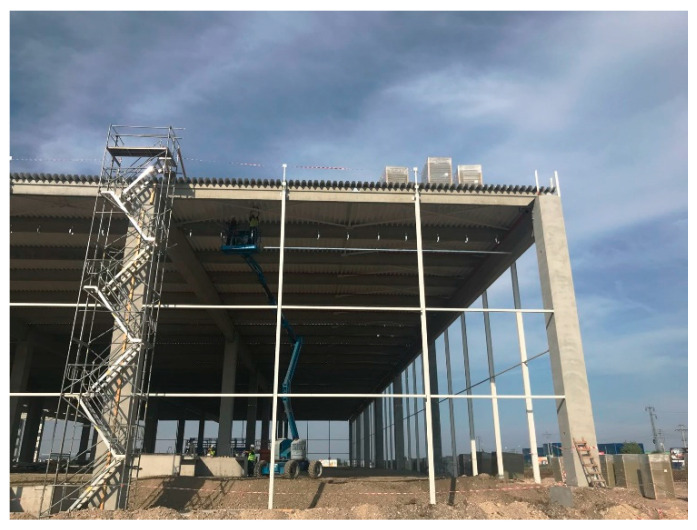
Load-bearing structure for a single-story warehouse on a site with severe seismic action, ground acceleration a_g_ = 0.35 g. Reproduced with permission from ©SDC Project [https://www.sdcproiect.ro/wp-content/uploads/2018/10/IMG_1887-08-06-18-11-30-1.jpg (accessed on 4 March 2024)].

**Figure 2 biomimetics-09-00165-f002:**
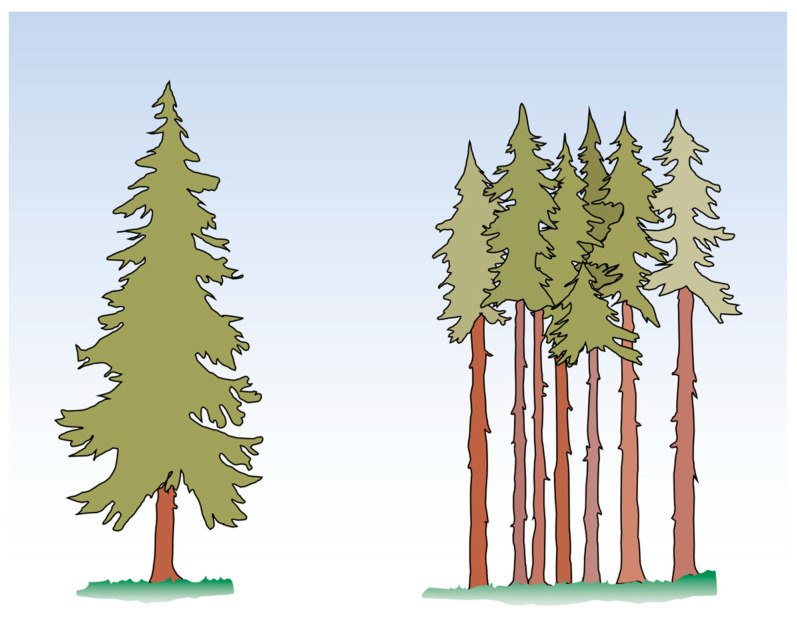
Coniferous forest trees (spruce species) as freestanding (**left**) versus densely growing (**right**). Adapted from ©Swedishwood [https://www.swedishwood.com/optimized/default/siteassets/1-trafakta/2-att-valja-tra/01/com/fristaende-gran-com.jpg/ (accessed on 4 March 2024)].

**Figure 3 biomimetics-09-00165-f003:**
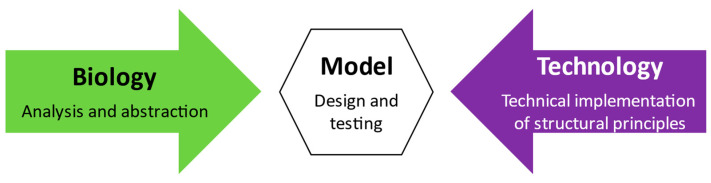
Interaction between the study of biology and technical implementation in biomimetics, based on [22].

**Figure 5 biomimetics-09-00165-f005:**
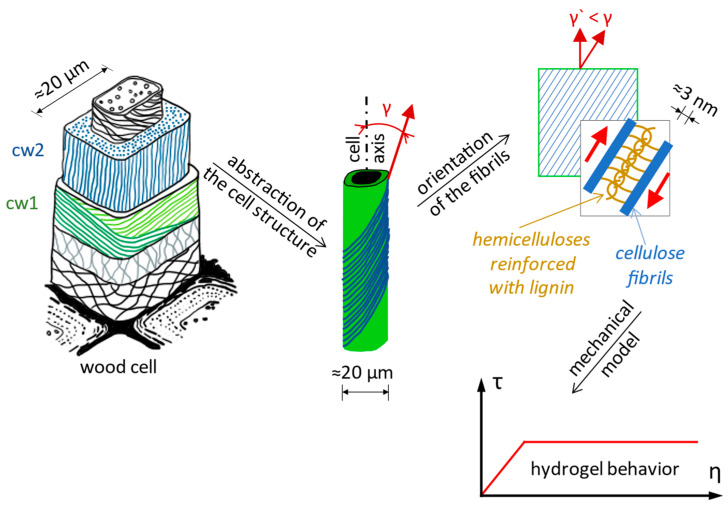
Cell wall structure in coniferous wood, representing the main cell walls cw1 and cw2, respectively, the middle lamella between the cells (**left** side), data from [68]. Geometry of cellulose microfibrils in the cell wall of coniferous wood and the angle made by the microfibrils with the longitudinal axis of the cells, γ, which decreases with increasing axial stress along the cell walls (in the **center**), data from [66,69,70]. The mechanical response of the cellulose fibrils embedded in the hemicellulose and lignin matrix, represented as a characteristic curve τ-η (shear stress–strain) (**right** corner), data from [66,69,71].

**Figure 6 biomimetics-09-00165-f006:**
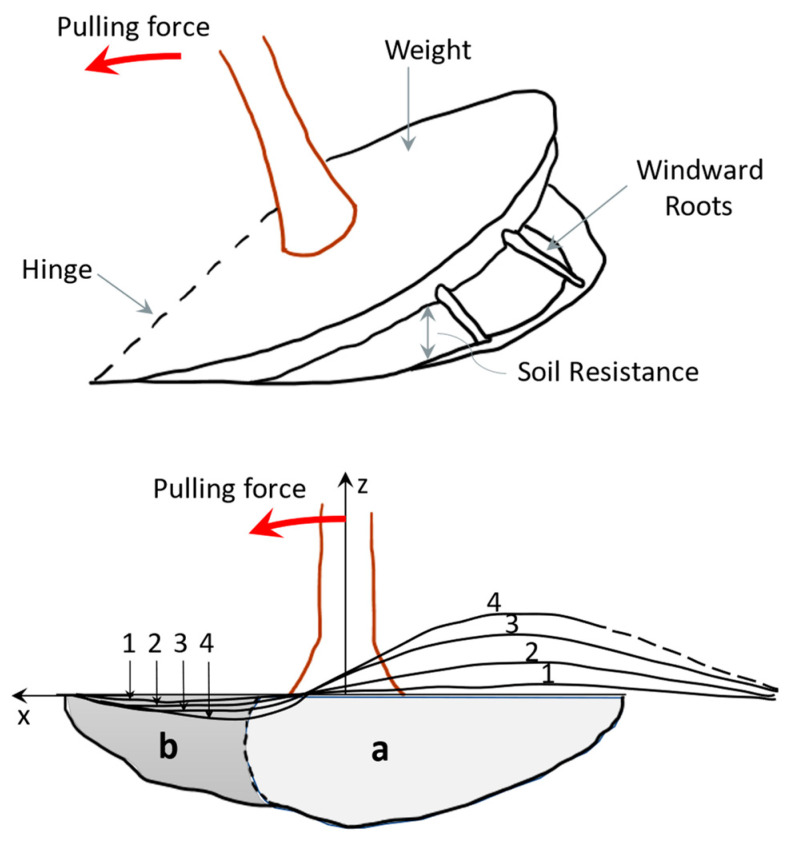
Diagrammatic view of a shallow-rooted tree, emphasizing the components of the anchorage (weight of the root–soil plate, soil resistance to uprooting, windward roots in tension, bending resistance of the root–soil system in the plastic hinge area) that resist the horizontal force acting on the stem. The curves 1 to 4 represent the deformation of the root-soil-plate as the pulling force grows. The area a is where the soil remains in the “elastic” domain, while in area b large deformation occurs, specific to a plastic zone. A Coutts concept adapted from [73,74,75].

**Figure 7 biomimetics-09-00165-f007:**
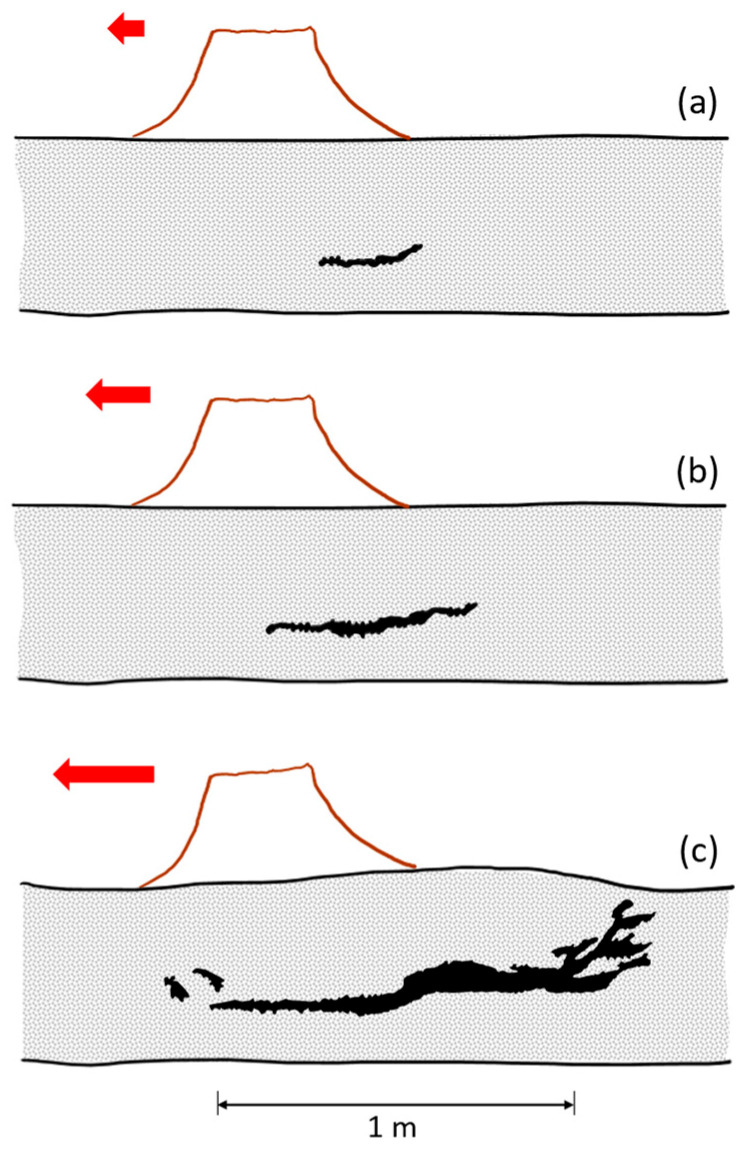
Sequences in root–soil system failure when a tree is pulled with an increasing horizontal force (red arrows). (**a**) Formation of crack close to stem base on windward side, (**b**) cracks extending windward and leeward, (**c**) appearance of cracks for the maximum uprooting turning moment, adapted from [73].

**Figure 8 biomimetics-09-00165-f008:**
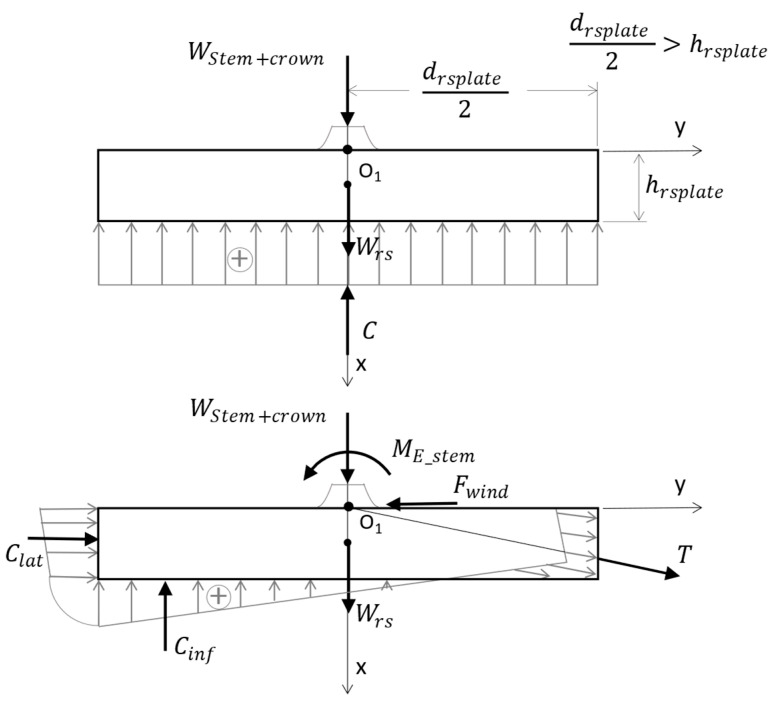
Distribution of reaction forces at the base of the foundation under the assumption of a centric compression (a compression with high eccentricity), a Grudnicki concept, adapted from [9]. The root–soil plate is considered to have an average diameter of d_rsplate_ and a depth of h_rsplate_.

**Figure 9 biomimetics-09-00165-f009:**
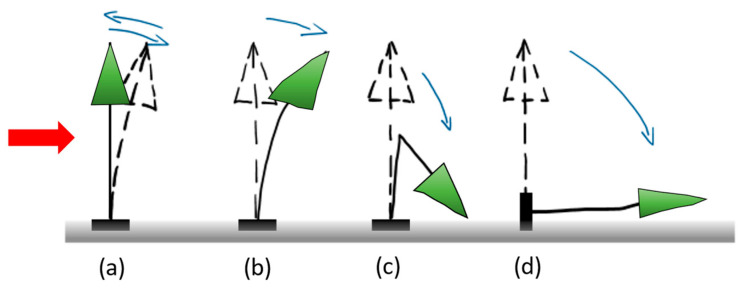
Shapes of tree deformability and instability due to wind action (the direction of wind is represented using the red arrow). (**a**) Bending and elastic buckling of the trunk, in which case the trunk returns to its original vertical position, (**b**) bending and plastic buckling of the trunk, in which case the stem remains deformed, (**c**) breakage of the trunk, and (**d**) overturning the tree by uprooting. Adapted from [7].

**Figure 11 biomimetics-09-00165-f011:**
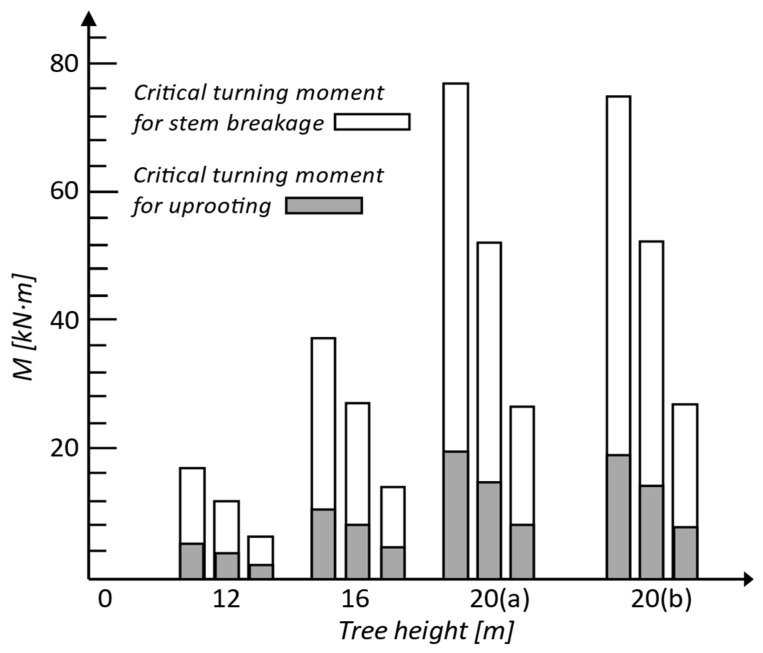
Turning moment needed to uproot versus turning moment needed to break the stem of Scots pine trees (crown-to-stem weight ratio (a) 0.3 and (b) 0.5). The mean ratio turning moment by gravity/turning moment by wind was less than 20%. Adapted from [86].

**Figure 12 biomimetics-09-00165-f012:**
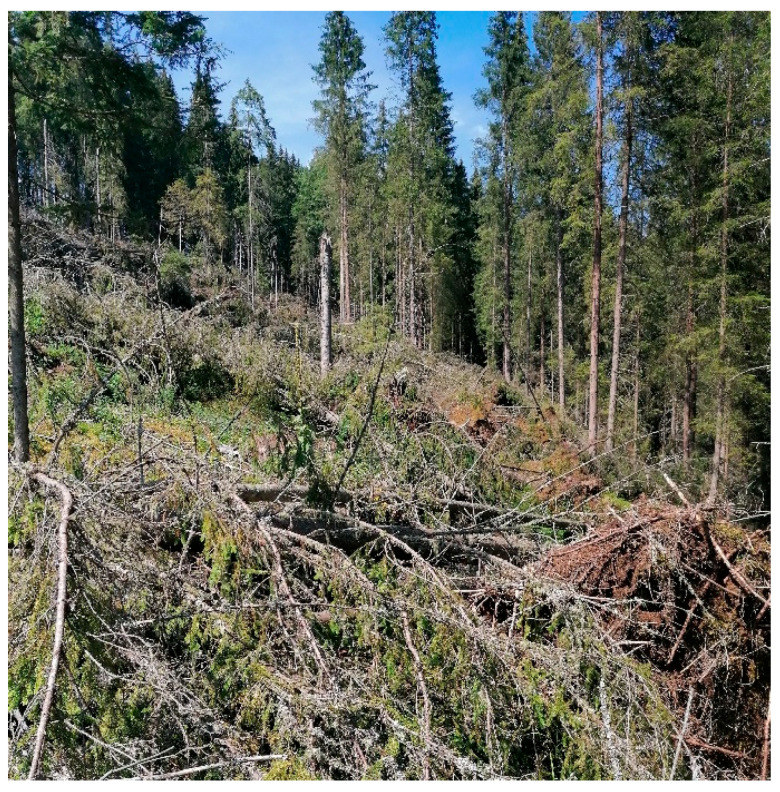
Coniferous trees blown down by the wind (near Bilbor, Călimani Mountains, Eastern Carpathians, Romania).

**Figure 13 biomimetics-09-00165-f013:**
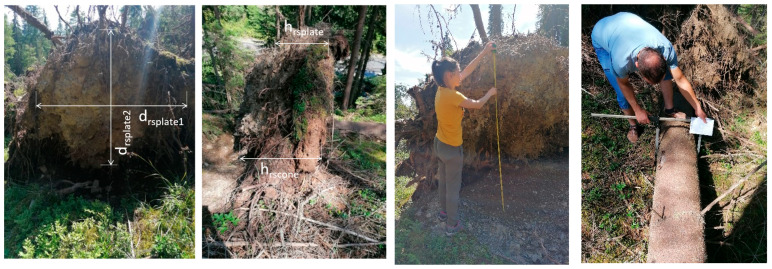
Measured parts of a root–soil plate: orthogonal diameters (d_rsplate1_ and d_rsplate2_) and depth (h_rsplate_ and h_rscone_) (near Bilbor, Călimani Mountains, Eastern Carpathians, Romania). Measurement on site of the root–soil plate size and the DBH.

**Figure 14 biomimetics-09-00165-f014:**
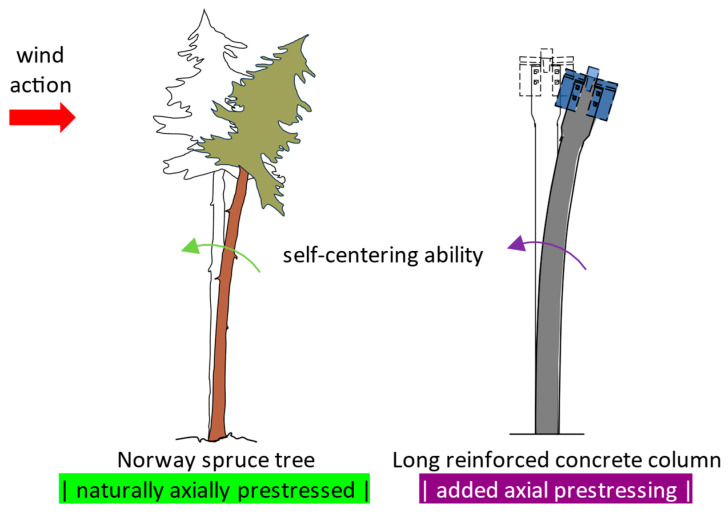
Deformed Norway spruce tree under wind loads, and deformed reinforced prestressed concrete column due to a horizontal point load at the top.

**Figure 15 biomimetics-09-00165-f015:**
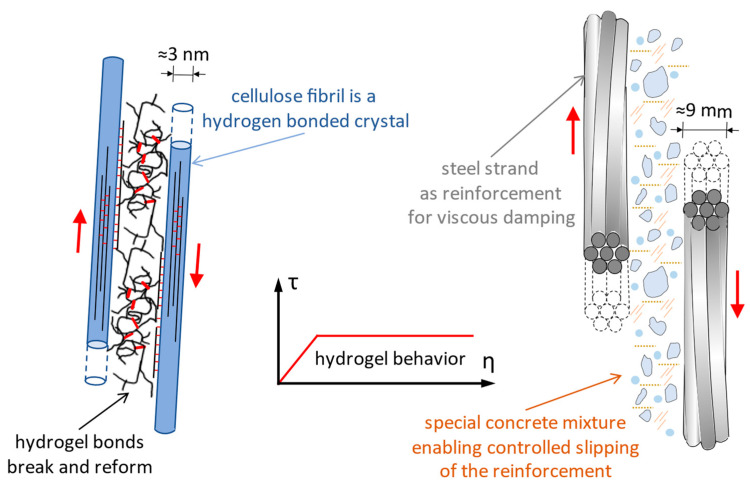
Deformation of cellulose fibril and of hemicellulose-lignin based matrix in cell walls of Norway spruce wood, and deformation of steel strands inside a concrete member when controlled bond slip of reinforcement is enabled. The red arrow represents the direction of the axial force inside of the fibril, respectively inside of the steel strand.

**Figure 16 biomimetics-09-00165-f016:**
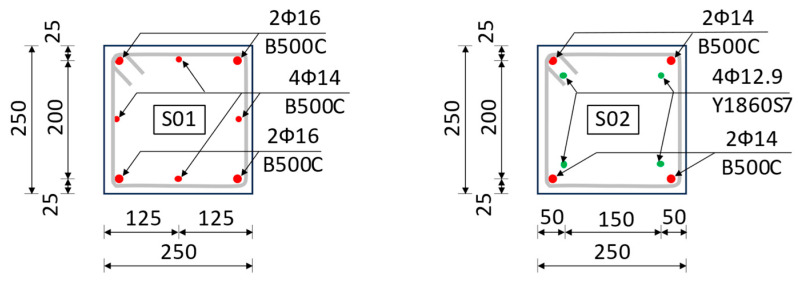
Cross-sections of S01 and S02, dimensions in mm. Adapted from [100].

**Figure 17 biomimetics-09-00165-f017:**
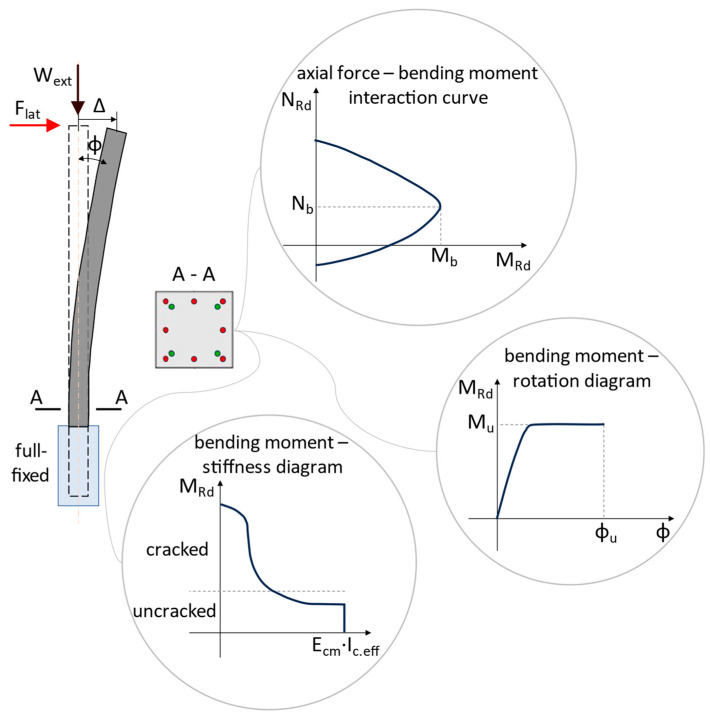
Sketch with test set-up and analyses of prestressed concrete columns. Green dots represent the active reinforcement, red dots the passive one. Adapted from [103].

**Figure 18 biomimetics-09-00165-f018:**
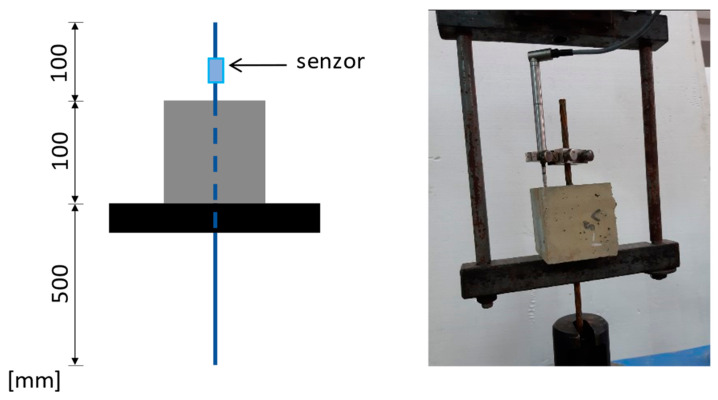
Sketch (not at scale) and photo with test set-up for reinforcement bond with concrete.

**Figure 19 biomimetics-09-00165-f019:**
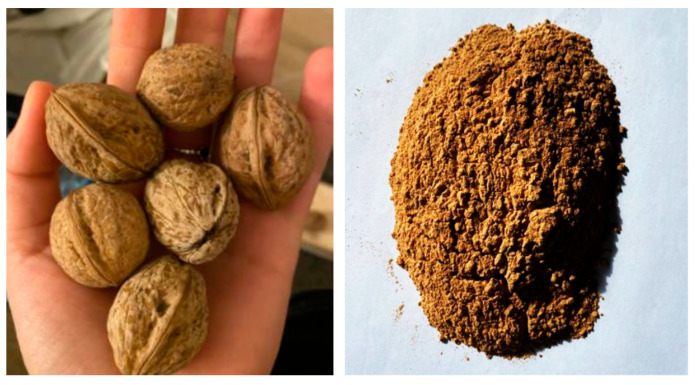
Walnut shell before and after fine grinding [105]. Reproduced with permission from Daria Rozian “Influența ligninei asupra rezistenței la compresiune a betonului și asupra aderenței armăturii în beton” (Master Dissertation), advisor: T.-N. Toader, UTCN, 2022.

**Figure 20 biomimetics-09-00165-f020:**
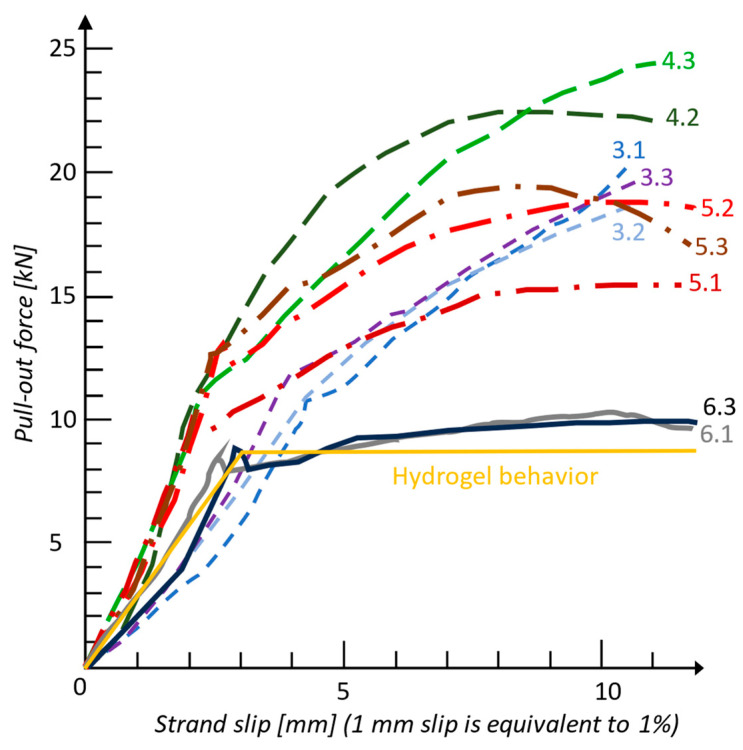
Force–slip curve for minimum 10% slip, obtained experimentally.

**Table 1 biomimetics-09-00165-t001:** Average value of green wood properties of some conifers, data from [72].

Genus/Species	Number of Trees Investigated	Modulus of Elasticity of Green Wood, E_g_ (GPa)	Density, ρ (kg/m^3^)	The Ratio of Crown Weight to Stem Weight, W_crown_/W_stem_
*Spruce* (*Picea* spp.)				
Norway spruce (Picea abies)	32	6.23	598	0.32
Sitka spruce (Picea sitchensis)	175	7.53	447	0.50
White spruce (Picea glauca)	6	7.40	466	0.34
*Pine* (*Pinus* spp.)				
Corsican pine (Pinus nigra)	57	8.70	657	0.34
Lodgepole pine (Pinus contorta)	40	6.90	487	0.33
Scots pine (Pinus sylvestris)	20	7.33	700	0.29
Red pine (Pinus resinosa)	300	8.80	410	0.22
*Douglas fir* (*Pseudotsuga* spp*.)*				
Douglas fir (Pseudotsuga menziesii)	17	9.83	583	0.16

**Table 2 biomimetics-09-00165-t002:** Descriptive characteristics of the 30 specimens of Norway spruce investigated in the Bilbor region.

No.	Height (m)	Diameter at Base (cm)	DBH (cm)	h_rsplate_ (cm)	d_rsplate1_ (m)	d_rsplate2_ (m)	M_R_rs_ (kN·m)	M_R_stem_ (kN·m)	M_R_stem_/M_R_rs_ (-)
1	29.0	41	36	25	3.4	2.7	34.2	140.2	4.1
2	29.0	42	36	30	3.5	1.8	37.2	140.2	3.8
3	28.5	38	32	27	2.3	1.6	16.3	98.4	6.0
4	30.0	43	38	34	3.4	2.4	57.3	164.8	2.9
5	30.5	45	39	29	3.2	2.0	33.5	178.2	5.3
6	31.0	51	42	30	3.5	2.4	46.1	222.6	4.8
7	30.5	50	41	26	3.4	2.5	34.7	207.0	6.0
8	30.0	43	39	29	3.2	2.1	34.8	178.2	5.1
9	27.5	36	32	20	2.0	1.4	6.8	98.4	14.5
10	29.0	40	38	24	3.0	2.3	23.8	164.8	6.9
11	29.5	41	38.5	30	3.2	2.4	41.6	171.4	4.1
12	28.0	39	37	29	3.0	1.8	28.5	152.2	5.3
13	27.0	38	34	26	2.8	1.7	20.2	118.1	5.9
14	28.0	37	32	28	2.1	1.7	16.7	98.4	5.9
15	31.0	42	38	30	2.4	2.1	26.8	164.8	6.1
16	30.0	43	38.5	30	3.1	2.2	37.2	171.4	4.6
17	29.0	36	34	28	2.2	1.7	17.6	118.1	6.7
18	28.5	41	36	30	3.0	1.8	30.5	140.2	4.6
19	29.0	40	36	27	3.3	2.5	36.1	140.2	3.9
20	30.5	41	39	34	3.5	2.6	63.3	178.2	2.8
21	30.0	48	40	30	3.4	2.6	47.7	192.3	4.0
22	31.0	50	42	26	3.6	2.7	39.5	222.6	5.6
23	27.0	35	32	25	2.2	1.6	13.3	98.4	7.4
24	27.0	37	34	28	2.7	2.0	25.5	118.1	4.6
25	30.0	40	37	32	3.2	2.2	44.0	152.2	3.5
26	28.0	36	33	30	3.1	2.0	34.5	108.0	3.1
27	30.0	50	42	34	3.4	2.2	53.4	222.6	4.2
28	30.0	48	40	27	3.1	2.6	34.9	192.3	5.5
29	28.5	42	37	29	2.8	2.1	29.7	152.2	5.1
30	30.5	51	42	32	3.5	2.5	54.3	222.6	4.1

Note: the height of the tree, the diameter of the stem at the base, the diameter of the stem at breast height, and the two diameters of the root–soil system (with elliptical shape in plane) were measured on site. The thickness of the root–soil plate was evaluated with on-site measurement. The ratio M_R_stem_/M_R_rs_ was calculated considering the following estimated values based on the on-site information correlated with the bibliographic resource [80], as follows: flexural strength of the green wood in stem f_fl,gw,stem_ = 30.6 MPa; modulus of elasticity of green wood = 6300 MPa; crown-to-stem weight ratio = 0.50; contribution of root–soil plate weight to total anchorage A_rsw_ = 20%; mean density of the fresh root–soil plate = 1500 kg/m^3^; mean density of green wood = 800 kg/m^3^. In Equation (1), for coniferous λ is considered 1.5, according to [72]. First natural frequency is calculated with Equation (1) and verified with Equation (2); in the end, same value of 0.10 Hz was obtained for all 30 specimens investigated.

**Table 3 biomimetics-09-00165-t003:** Mechanical properties of used materials, data from [100].

	Modulus of Elasticity MPa	Characteristic Strength (f_ck_, f_pk_, f_yk_) MPa	Ultimate Strain (ε_cu_, ε_pu_, ε_su_) [–]
Concrete C60/75	39,000	60	0.3%
Prestressing steel Y1860S7	199,000	1860	2.2%
Reinforcement B500C	205,000	500	7.5%

**Table 4 biomimetics-09-00165-t004:** Results of the pull-out tests.

Specimen		Concrete Grade	Maximum Value of Pull-Out Force kN	Average Value for Maximum Force kN
V3	V3–1	C32/40	21.39	19.88
	V3–2		21.9	
	V3–3		16.35	
V4	V4–1	C35/45	16.98	17.43
	V4–2		16.92	
	V4–3		18.39	
V5	V5–1	C35/45	15.33	17.2
	V5–2		18.81	
	V5–3		17.46	
V6	V6–1	C35/45	14.25	12.06
	V6–2		N/A	
	V6–3		9.87	

**Table 5 biomimetics-09-00165-t005:** Chemical composition of peanut shell and walnut shell.

Compound	Walnut Shell [106]	Peanut Shell [107]
Ash	3.4%	3.8%
Lignin	50.3%	36.1%
Hemicellulose	22.4%	5.6%
Cellulose	23.9%	44.8%

**Table 6 biomimetics-09-00165-t006:** Concrete ingredients used to obtain 3.3 L of fresh concrete (grade C35/45) to cast specimen V6.

Ingredient	Type	Amount
CEM I 52.5R	Cement	1155 g
Water	Water	533 g
Dry aggregates	Source: river	
Sand	0–2 mm	2767 g
Fine gravel	2–8 mm	1320 g
Medium gravel	8–16 mm	1921 g
Sika Plastiment BV 440	Plasticizer (lignosulfonate based)	13.3 g
Walnut (Juglans regia) shell powder	0.063–0.125 mm	11.5 g

## Data Availability

The data that support the findings of this study are available from the corresponding author upon reasonable request.

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
