# Peer review of "Coniferous Trees as Bioinspiration for Designing Long Reinforced Prestressed Concrete Columns"

_biomimetics, 2024, doi:10.3390/biomimetics9030165_

Round 1
Reviewer 1 Report
Comments and Suggestions for Authors
This article managed to identify the relevant mechanical, physical, dynamic and morphological properties to verify the potential of coniferous trees as a biological role model in the design of long reinforced and prestressed concrete columns. The research inquiry is precisely delineated, the dataset is thorough, and the workload is extensive. The analyses of the structural and mechanical properties of coniferous trees are methodologically sound. I recommend accepting this manuscript. There are only two aspects that could be further optimized in the article.
1. The overall structure of the article appears somewhat loose. For example, the section “2. Analysis and Abstraction of the Biological Model” encompasses topics such as Damping, Morphology, Plastic deformation, Young's modulus, Root system, and Failure under wind action. However, it is challenging to delve deeply into each subtopic and establish connections between them. Additionally, for aspects like Morphology, there seems to be a lack of actual testing data.
In the section “3. Design Methodology transfer from Coniferous Trees to Load-bearing Structures,” it remains unclear which two structural or component features of coniferous trees were specifically biomimicked to enhance the tensile strength of concrete columns. Further clarification is needed to elucidate the aspects of coniferous trees that were incorporated into the design methodology for improving tensile strength.
That is, the article does not need to be exhaustive but should focus on elucidating relevant points of the work.
2. Due to the issue mentioned above, there is a challenge in aligning the third and second sections, resulting in a somewhat rigid depiction of the biomimicry aspect. Therefore, it is essential to establish coherence between the third and second sections. The focus should be on depicting and elucidating the model, mechanism, data, and results of the real work in this study.
Reviewer 2 Report
Comments and Suggestions for Authors
The work is interesting, and the literature review is extensive, but I think it has little relationship with the experimental tests in point 4.2.
The various choices to arrive at the options adopted, in point 4.1, should be corroborated with the literature survey.
It would enrich the work if some images in the supplementary material (S06, S10, S14) were included.
The test referring to Figure 18 and the results should be better explained with a deeper analysis, given that, in my opinion, they are more relevant to the bibliographic analysis carried out.
Comments on the Quality of English LanguageThe quality of English is good.
Round 2
Reviewer 2 Report
Comments and Suggestions for Authors
The authors made the necessary revisions.